# FOURIER STOCHASTIC BACKPROPAGATION

## ABSTRACT

Backpropagating gradients through random variables is at the heart of numerous machine learning applications. In this paper, we present a general framework for deriving stochastic backpropagation rules for any distribution, discrete or continuous. Our approach exploits the link between the characteristic function and the Fourier transform, to transport the derivatives from the parameters of the distribution to the random variable. Our method generalizes previously known estimators, and results in new estimators for the gamma, beta, Dirichlet and Laplace distributions. Furthermore, we show that the classical deterministic backproapagation rule and the discrete random variable case, can also be interpreted through stochastic backpropagation.

## 1 INTRODUCTION

Deep neural networks with stochastic hidden layers have become crucial in multiple domains, such as generative modeling (Kingma & Welling, 2013; Rezende et al., 2014; Mnih & Gregor, 2014), deep reinforcement learning (Sutton et al., 2000), and attention mechanisms (Mnih et al., 2014). The difficulty encountered in training such models arises in the computation of gradients for functions of the form $\mathcal{L}(\theta) := \mathbb{E}_{\mathbf{z} \sim p_\theta} [f(\mathbf{z})]$ with respect to the parameters $\theta$, thus needing to backpropagate the gradient through the random variable $\mathbf{z}$. One of the first and most used methods is the score function or reinforce method (Glynn, 1989; Williams, 1992), that requires the computation and estimation of the derivative of the log probability function. For high dimensional applications however, it has been noted that reinforce gradients have high variance, making the training process unstable (Rezende et al., 2014).

Recently, significant progress has been made in tackling the variance problem. The first class of approaches dealing with continuous random variables are reparameterization tricks. In that case a standardization function is introduced, that separates the stochasticity from the dependency on the parameters $\theta$. Thus being able to transport the derivative inside the expectation and sample from a fixed distribution, resulting in low variance gradient (Kingma & Welling, 2013; Rezende et al., 2014; Titsias & Lázaro-Gredilla, 2014; Ruiz et al., 2016; Naesseth et al., 2017; Figurnov et al., 2018). The second class of approaches concerns discrete random variables, for which a direct reparameterization is not known. The first solution uses the score function gradient with control variate methods to reduce its variance (Mnih & Gregor, 2014; Gu et al., 2016). The second consists in introducing a continuous relaxation admitting a reparameterization trick of the discrete random variable, thus being able to backpropagate low-variance reparameterized gradients by sampling from the concrete distribution (Jang et al., 2016; Maddison et al., 2016; Tucker et al., 2017; Grathwohl et al., 2018).

Although recent developments have advanced the state-of-the-art in terms of variance reduction and performance, stochastic backpropagation (i.e computing gradients through random variables) still lacks theoretical foundation. In particular, the following questions remain open: How to develop stochastic backpropagation rules, where the derivative is transferred explicitly to the function $f$ for a broader range of distributions? And can the discrete and deterministic cases be interpreted in the sense of stochastic backpropagation? In this paper, we provide a new method to address these questions, and our main contributions are the following:

- We present a theoretical framework based on the link between the multivariate Fourier transform and the characteristic function, that provides a standard method for deriving stochastic backpropagation rules, for **any** distribution discrete or continuous.

- We show that deterministic backpropagation can be interpreted as a special case of stochastic backpropagation, where the probability distribution $p_\theta$ is a Dirac delta distribution, and that the discrete case can also be interpreted as backpropagating a discrete derivative.

- We generalize previously known estimators, and provide new stochastic backpropagation rules for the special cases of the Laplace, gamma, beta, and Dirichlet distributions.

- We demonstrate experimentally that the resulting new estimators are competitive with state-of-the art methods on simple tasks.

## 2    BACKGROUND & PRELIMINARIES

Let $(E, \lambda)$ be a $d$-dimensional measure space equipped with the standard inner product, and $f$ be a square summable positive real valued function on $E$, that is, $f \colon E \to \mathbb{R}_+$, with $\int_E |f(z)|^2 \lambda(dz) < \infty$. Let $p_\theta$ be an arbitrary parameterized probability density on the space $E$. We denote by $\varphi_\theta$ its characteristic function, defined as: $\varphi_\theta(\omega) := \mathbb{E}_{\mathbf{z} \sim p_\theta}[e^{i\omega^T \mathbf{z}}]$. We denote by $\hat{f}$ the Fourier transform of the function $f$ defined as:

$$\hat{f}(\omega) := \mathcal{F}\{f\}(\omega) = \int_E f(z)e^{-i\omega^T z} \lambda(dz). \tag{1}$$

The inverse Fourier transform is given in this case by:

$$f(z) := \mathcal{F}^{-1}\{\hat{f}\}(z) = \int_{\mathbb{R}^d} \hat{f}(\omega)e^{i\omega^T z} \mu(d\omega), \tag{2}$$

where $\mu(d\omega)$ represents the measure in the Fourier domain. In this paper we treat the cases where $E = \mathbb{R}^d$ for which $\mu(d\omega) = \frac{d\omega}{(2\pi)^d}$, and the case where $E$ is a discrete set, for which the measure $\mu$ is defined as: $\mu(d\omega) = \mathbb{1}[\omega \in [-\pi, \pi]^d] \frac{d\omega}{(2\pi)^d}$. Throughout the paper, we reserve the letter $i$ to denote the imaginary unit: $i^2 = -1$. To denote higher order derivatives of the function $f$, we use the multi-index notation (Saint Raymond, 2018). For a multi-index $n = (n_1, ..., n_d) \in \mathbb{N}^d$, we define:

$$\partial_z^n := \frac{\partial^{|n|}}{\partial z_1^{n_1} ... \partial z_d^{n_d}} \quad \text{where} \quad |n| = \sum_{j=1}^d n_j \quad \text{and} \quad \omega^n := \prod_{j=1}^d \omega_j^{n_j}.$$

To clarify the multi-index notation, let us consider the example where $d = 3$, and $n = (1, 0, 2)$, in this case:

$$\partial_z^n = \frac{\partial^3}{\partial z_1 \partial z_3^2} \quad \text{and,} \quad \omega^n = \omega_1 \omega_3^2.$$

The objective is to derive stochastic backpropagation rules, similar to that of (Rezende et al., 2014), for functions of the form: $\mathcal{L}(\theta) := \mathbb{E}_{\mathbf{z} \sim p_\theta}[f(\mathbf{z})]$, for any arbitrary distribution $p_\theta$, discrete or continuous.

## 3    FOURIER STOCHASTIC BACKPROPAGATION

Stochastic backpropagation rules similar to that of (Rezende et al., 2014) can in fact be derived for any continuous distribution, under certain conditions on the characteristic function. In the following theorem we present the main result of our paper concerning the derivation of Fourier stochastic backpropagation rules.

**Theorem 1.** *(Continuous Stochastic Backpropagation) Let $f \in \mathcal{C}^\infty(\mathbb{R}^d, \mathbb{R}_+)$, under the condition that $\nabla_\theta \log \varphi_\theta$ is a holomorphic function of $i\omega$, then there exists a unique $\{a_n(\theta)\}_{n \in \mathbb{N}^d} \in \mathbb{R}$ such that:*

$$\nabla_\theta \mathcal{L} = \sum_{|n| \geq 0} a_n(\theta) \mathbb{E}_{\mathbf{z} \sim p_\theta}\left[\partial_z^n f(\mathbf{z})\right]. \tag{3}$$

*Where $\{a_n(\theta)\}_{n \in \mathbb{N}^d}$ are the Taylor expansion coefficients of $\nabla_\theta \log \varphi_\theta(\omega)$:*

$$\nabla_\theta \log \varphi_\theta(\omega) = \sum_{|n| \geq 0} a_n(\theta)(i\omega)^n. \tag{4}$$

*Proof.* Let us rewrite $\mathcal{L}$ in terms of $\hat{f}$:

$$
\begin{aligned}
\mathcal{L}(\theta) &= \int p_\theta(z) f(z) \lambda(dz) \\
&= \int p_\theta(z) \mathcal{F}^{-1}[\hat{f}](z) \lambda(dz) \\
&= \int_{\mathbb{R}^d} \hat{f}(\omega) \int_E p_\theta(z) e^{i\omega^T z} \lambda(dz) \mu(d\omega) \quad \text{Fubini's theorem} \\
&= \int_{\mathbb{R}^d} \hat{f}(\omega) \varphi_\theta(\omega) \mu(d\omega).
\end{aligned}
\tag{5}
$$

By introducing the derivative under the integral sign, and using the reinforce trick (Williams, 1992) applied to $\varphi_\theta$, where $\nabla_\theta \varphi_\theta(\omega) = \varphi_\theta(\omega) \nabla_\theta \log \varphi_\theta(\omega)$, equation 5 becomes:

$$
\nabla_\theta \mathcal{L} = \int_{\mathbb{R}^d} \hat{f}(\omega) \varphi_\theta(\omega) \nabla_\theta \log \varphi_\theta(\omega) \mu(d\omega).
\tag{6}
$$

Under analyticity conditions of the gradient of the log characteristic function, we can expand the gradient term $\nabla_\theta \log \varphi_\theta(\omega)$, in terms of Taylor series around zero as:

$$
\nabla_\theta \log \varphi_\theta(\omega) = \sum_{|n| \geq 0} a_n(\theta)(i\omega)^n.
\tag{7}
$$

Putting everything together, and replacing the characteristic function by its expression, the gradient of $\mathcal{L}$ becomes:

$$
\nabla_\theta \mathcal{L} = \int_{\mathbb{R}^d} \hat{f}(\omega) \int_E p_\theta(z) e^{i\omega^T z} \sum_{|n| \geq 0} a_n(\theta)(i\omega)^n \mu(d\omega) \lambda(dz).
\tag{8}
$$

By rearranging the sums using Fubini's theorem a second time, we obtain the following expression for the gradient:

$$
\begin{aligned}
\nabla_\theta \mathcal{L} &= \mathbb{E}_{\mathbf{z} \sim p_\theta} \left[ \mathcal{F}^{-1} \left\{ \omega \mapsto \sum_{|n| \geq 0} a_n(\theta)(i\omega)^n \hat{f}(\omega) \right\} (\mathbf{z}) \right] \\
&= \sum_{|n| \geq 0} a_n(\theta) \mathbb{E}_{\mathbf{z} \sim p_\theta} \left[ \mathcal{F}^{-1} \left\{ \omega \mapsto (i\omega)^n \hat{f}(\omega) \right\} (\mathbf{z}) \right] \\
&= \sum_{|n| \geq 0} a_n(\theta) \mathbb{E}_{\mathbf{z} \sim p_\theta} \left[ \partial_z^n f(\mathbf{z}) \right].
\end{aligned}
\tag{9}
$$

Q.E.D

Identically, we can follow the same procedure for discrete random variables. We suppose that $p_\theta$ factorizes over disjoint cliques of the dependency graph, where each dimension $\mathbf{z}_j$ takes values in a discrete space $\mathcal{V}al(z_j)$. In theorem 2 we derive the result concerning the discrete case.

**Theorem 2.** *(Discrete Stochastic Backpropagation) Let E be a discrete space: $E = \prod_{j=1}^d \mathcal{V}al(z_j)$, and $\mathcal{C}$ the set of disjoint cliques of the dependency graph over z, that is,*

$$
p_\theta(z) = \prod_{c \in \mathcal{C}} p_\theta(z_c)
$$

*then,*

$$
\nabla_\theta \mathcal{L} = \sum_{c \in \mathcal{C}} \sum_{z_c \neq z_c^*} \nabla_\theta p_\theta(z_c) \mathbb{E}_{\mathbf{z}_{-c} \sim p_\theta} \left[ \mathbf{D} f(\mathbf{z}_{-c}, z_c) \right].
\tag{10}
$$

*Where:*

- $z_c^*$*: represents the normalizing assignment* $p_\theta(z_c^*) = 1 - \sum_{z_c \neq z_c^*} p_\theta(z_c)$.

$$\bullet \ \mathbf{D}f(\mathbf{z}_{-c}, z_c) := f(\mathbf{z}_{-c}, z_c) - f(\mathbf{z}_{-c}, z_c^*). \tag{11}$$

*Proof.* The characteristic function for the factored distribution is given by:

$$\varphi_\theta(\omega) = \prod_{c \in \mathcal{C}} \varphi_\theta^{(c)}(\omega_c), \quad \varphi_\theta^{(c)}(\omega_c) = \sum_{z_c \neq z_c^*} p_\theta(z_c) e^{i\omega_c^T z_c} + \left(1 - \sum_{z_c \neq z_c^*} p_\theta(z_c)\right) e^{i\omega_c^T z_c^*}. \tag{12}$$

Thus the gradient of the log characteristic function becomes:

$$\nabla_\theta \log \varphi_\theta^{(c)}(\omega_c) = \sum_{z_c \neq z_c^*} \nabla_\theta p_\theta(z_c) \left[ \frac{e^{i\omega_c^T z_c} - e^{i\omega_c^T z_c^*}}{\varphi_\theta^{(c)}(\omega_c)} \right]. \tag{13}$$

By plugging this expression to equation 6, we obtain:

$$\begin{aligned}
\nabla_\theta \mathcal{L} &= \sum_{c \in \mathcal{C}} \sum_{z_c \neq z_c^*} \nabla_\theta p_\theta(z_c) \int \prod_{c' \neq c} \varphi_\theta^{(c')}(\omega_{c'}) \left[ e^{i\omega_c^T z_c} - e^{i\omega_c^T z_c^*} \right] \hat{f}(\omega) \mu(d\omega) \\
&= \sum_{c \in \mathcal{C}} \sum_{z_c \neq z_c^*} \nabla_\theta p_\theta(z_c) \mathbb{E}_{\mathbf{z}_{-c} \sim p_\theta} \left[ \mathbf{D}f(\mathbf{z}_{-c}, z_c) \right].
\end{aligned} \tag{14}$$

Q.E.D

The estimator of equation 10 has been derived in the literature through Rao–Blackwellization of the score function gradient, and it has been known under different names (Titsias & Lázaro-Gredilla, 2015; Asadi et al., 2017; Cong et al., 2019). Theorem 2 shows that the discrete case can also be seen as backpropagating a derivative of the function $f$, in this case a discrete derivative given by equation 11.

## 4 APPLICATIONS OF FOURIER STOCHASTIC BACKPROPAGATION

Following from the previous section, we derive the stochastic backpropagation estimators for certain commonly used distributions.

**The multivariate Gaussian distribution:** In this case $p_\theta(z) = \mathcal{N}(z; \mu_\theta, \Sigma_\theta)$. The log characteristic function is given by: $\log \varphi_\theta(\omega) = i\mu_\theta^T \omega + \frac{1}{2}\text{Tr}\left[\Sigma_\theta i^2 \omega \omega^T\right]$. Thus by applying theorem 1, we recover the stochastic backpropagation rule of (Rezende et al., 2014):

$$\nabla_\theta \mathcal{L} = \mathbb{E}_{\mathbf{z} \sim p_\theta} \left\{ \left( \frac{\partial \mu_\theta}{\partial \theta} \right)^T \nabla_z f(\mathbf{z}) + \frac{1}{2}\text{Tr}\left[ \left( \frac{\partial \Sigma_\theta}{\partial \theta} \right) \nabla_z^2 f(\mathbf{z}) \right] \right\}, \tag{15}$$

where, $\nabla_z$ and $\nabla_z^2$, represent the gradient and hessian operators.

**The multivariate Dirac distribution:** $p_\theta(z) = \delta_{a_\theta}(z)$, the log characteristic function of the Dirac distribution is given by: $\log \varphi_\theta(\omega) = i\omega^T a_\theta$. Thus the stochastic backpropagation rule of the Dirac is given by:

$$\nabla_\theta \mathcal{L} = \left( \frac{\partial a_\theta}{\partial \theta} \right)^T \mathbb{E}_{\mathbf{z} \sim \delta_{a_\theta}} \left[ \nabla_z f(\mathbf{z}) \right] = \left( \frac{\partial a_\theta}{\partial \theta} \right)^T \nabla_z f(a_\theta), \tag{16}$$

resulting in the classical backpropagation rule. In other words, the deterministic backpropagation rule is a special case of stochastic backpropagation where the distribution is a Dirac delta distribution. This result provides a link between probabilistic graphical models and classical neural networks. We investigate this link further in Appendix A.

**The multivariate Bernoulli:** $p_\theta(z) = \prod_{j=1}^d \mathcal{B}(z_j; \pi_\theta^{(j)})$, where $\pi_\theta^{(j)} = \mathbb{P}[z_j = 1]$. By applying theorem 2, we obtain the local expectation gradient of (Titsias & Lázaro-Gredilla, 2015):

$$\nabla_\theta \mathcal{L} = \sum_{j=1}^d \frac{\partial \pi_\theta^{(j)}}{\partial \theta} \mathbb{E}_{\mathbf{z}_{-j} \sim p_\theta} \left[ f(\mathbf{z}_{-j}, 1) - f(\mathbf{z}_{-j}, 0) \right]. \tag{17}$$

**The multivariate categorical:** $p_\theta(z) = \prod_{j=1}^d \mathrm{cat}(z_j; \pi_\theta^{(j)})$, where the dimensions are independent and take values in the set $\{1, ..., K\}$. Similarly to the Bernoulli case, we obtain the following stochastic backpropagation rule:

$$\nabla_\theta \mathcal{L} = \sum_{j=1}^d \sum_{k=1}^{K-1} \frac{\partial \pi_{\theta_k}^{(j)}}{\partial \theta} \mathbb{E}_{\mathbf{z}_{-j} \sim p_\theta} \left[ \mathbf{D} f(\mathbf{z}_{-j}, k) \right]. \tag{18}$$

**The Laplace distribution:** $p_\theta(z) = L(z; \mu_\theta, b_\theta)$, in this case the log characteristic function is the following: $\log \varphi_\theta(\omega) = i\mu_\theta \omega - \log(1 + b_\theta^2 \omega^2)$, using the Taylor series expansion for the function $x \mapsto \frac{1}{1-x}$, we get the following stochastic backpropagation rule for the Laplace distribution:

$$\nabla_\theta \mathcal{L} = \frac{\partial \mu_\theta}{\partial \theta} \mathbb{E}_\mathbf{z} \left[ \frac{df}{dz}(\mathbf{z}) \right] + \frac{1}{b_\theta^2} \frac{\partial b_\theta^2}{\partial \theta} \sum_{n=1}^\infty b_\theta^{2n} \mathbb{E}_\mathbf{z} \left[ \frac{d^{2n} f}{dz^{2n}}(\mathbf{z}) \right]. \tag{19}$$

**The gamma distribution:** $p_\theta(z) = \Gamma(z; k_\theta, \mu_\theta)$, the log characteristic function of the Gamma distribution is given by: $\log \varphi_\theta(\omega) = -k_\theta \log(1 - i\mu_\theta \omega)$. By expanding it using Taylor series of the logarithm function, we obtain the following stochastic backpropagation rule:

$$\nabla_\theta \mathcal{L} = \sum_{n=1}^\infty \left[ \frac{1}{n} \frac{\partial k_\theta}{\partial \theta} + \frac{k_\theta}{\mu_\theta} \frac{\partial \mu_\theta}{\partial \theta} \right] \mu_\theta^n \mathbb{E}_{\mathbf{z} \sim p_\theta} \left[ \frac{d^n f}{dz^n}(\mathbf{z}) \right]. \tag{20}$$

The estimator of equation 20 gives a stochastic backpropagation rule for the gamma distribution and, hence also applies by extension to the special cases of the exponential, Erlang, and chi-squared distributions.

**The beta distribution:** $p_\theta(z) = \mathrm{Beta}(z; \alpha_\theta, \beta_\theta)$, in this case the characteristic function is the confluent hypergeometric function: $\varphi_\theta(\omega) = {}_1F_1(\alpha_\theta; \alpha_\theta + \beta_\theta; i\omega)$. A series expansion of the gradient of the log of this function is not trivial to derive. However, we can use the parameterization linking the gamma and beta distributions to derive a stochastic backpropagation rule. Indeed, if $\zeta_1 \sim \Gamma(\alpha_\theta, 1)$ and $\zeta_2 \sim \Gamma(\beta_\theta, 1)$, then $\mathbf{z} = g(\zeta_1, \zeta_2) = \frac{\zeta_1}{\zeta_1 + \zeta_2} \sim \mathrm{Beta}(\alpha_\theta, \beta_\theta)$. By substituting in the gamma stochastic backpropagation rule, we obtain:

$$\nabla_\theta \mathcal{L} = \sum_{n=1}^\infty \frac{1}{n} \left\{ \frac{\partial \alpha_\theta}{\partial \theta} \mathbb{E}_{\zeta_1, \zeta_2} \left[ \frac{\partial^n f}{\partial \zeta_1^n} \left( \frac{\zeta_1}{\zeta_1 + \zeta_2} \right) \right] + \frac{\partial \beta_\theta}{\partial \theta} \mathbb{E}_{\zeta_1, \zeta_2} \left[ \frac{\partial^n f}{\partial \zeta_2^n} \left( \frac{\zeta_1}{\zeta_1 + \zeta_2} \right) \right] \right\}. \tag{21}$$

**The Dirichlet distribution:** $p_\theta(z) = \mathrm{Dir}(z; K, \alpha_\theta)$, following the same procedure, as for the beta distribution and using the following parameterization: $\mathbf{z}_k = \frac{\zeta_k}{\sum_{j=1}^K \zeta_j}$ with, $\zeta_k \sim \Gamma(\alpha_\theta^{(k)}, 1)$, we obtain:

$$\nabla_\theta \mathcal{L} = \sum_{n=1}^\infty \frac{1}{n} \left\{ \sum_{k=1}^K \frac{\partial \alpha_\theta^{(k)}}{\partial \theta} \mathbb{E}_{\zeta_j \forall j} \left[ \frac{\partial^n f}{\partial \zeta_k^n} \left( \frac{\zeta_1}{\sum_{j=1}^K \zeta_j}, ..., \frac{\zeta_K}{\sum_{j=1}^K \zeta_j} \right) \right] \right\}. \tag{22}$$

## 5 TRACTABLE CASES & APPROXIMATIONS OF FOURIER STOCHASTIC BACKPROPAGATION

The Fourier stochastic backpropagation gradient as presented in previous sections presents two major computational bottlenecks for non-trivial distributions. The first is the computation of infinite series, and the second is evaluating higher order derivatives of the function $f$. Depending on the application, the function $f$ could be chosen in order to bypass the computational bottlenecks. A trivial example, is if the higher order derivatives of the function $f$ vanish at a certain order: $\partial_z^n f = 0$. Another example, is the exponential function $f(z) = \exp(\epsilon^T z)$. From the fact that it obeys the following partial differential equation $\frac{\partial f}{\partial z_j}(z) = \epsilon_j f(z)$, one can deduce that the stochastic backpropagation rule reduces in this case to:

$$\nabla_\theta \mathcal{L} = \nabla_\theta \log \varphi_\theta \left( \frac{\epsilon}{i} \right) \mathbb{E}_{\mathbf{z} \sim p_\theta} \left[ f(\mathbf{z}) \right] \tag{23}$$

In most real world applications however, the infinite sum will not often reduce to a tractable expression such as that of the exponential. An example of this case is the evidence lower bound of a generative

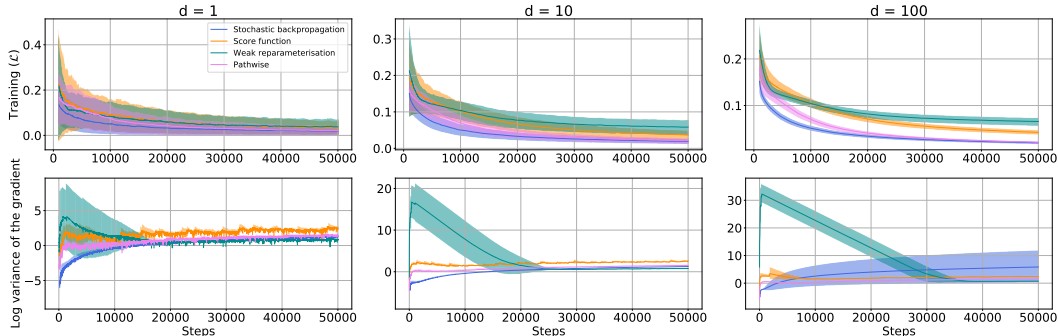

Figure 1: Training loss and log variance of the gradients for the different estimators for $f(z) = \sum_{j=1}^{d}(z_j - \epsilon)^2$ for $d \in \{1, 10, 100\}$.

model with Bernoulli observations. In this case, the natural solution is to truncate the sum up to a finite order. The assumption (although it might be wrong), is that the components associated to higher frequencies of the spectrum of the gradient of the log characteristic function, do not contribute as much. And by analogy to the signal processing field, we apply a Low-pass filter to eliminate them. In this case the gradient of the log characteristic function of equation 7 becomes:

$$\nabla_\theta \log \varphi_\theta(\omega) = \sum_{n \leq N} a_n(\theta)(i\omega)^n + o((i\omega)^N). \tag{24}$$

## 6 EXPERIMENTS

In our experimental evaluations, we test the stochastic backpropagation estimators of equations 19 and 20 for the gamma and Laplace distributions. In the case of the gamma estimator, we use toy examples where we can derive exact stochastic backpropagation rules without truncating the infinite sum. As for the Laplace stochastic backpropagation rule, we test the estimator in the case of Bayesian logistic regression with Laplacian priors and variational posteriors on the weights. We compare our estimators with the pathwise (Jankowiak & Karaletsos, 2019; Jankowiak & Obermeyer, 2018), and score function estimators, in addition to the weak reparameterization estimator in the gamma case (Ruiz et al., 2016). We do not use control variates in our setup, the goal is to verify the exactness of the proposed infinite series estimators and how they compare to current state-of-the-art methods in simple settings. In all our experiments, we use the Adam optimizer to update the weights (Kingma & Ba, 2014), with a standard learning rate of $10^{-3}$. In all the curves, we report the mean and standard deviation for all the metrics considered over 5 iterations.

### 6.1 TOY PROBLEMS

In the toy problem setting, we test the gamma stochastic backpropagation rule following the same procedure as (Mohamed et al., 2019). we consider the following cases:

**Toy problem 1**: $\mathcal{L}(\theta) = \mathbb{E}_{\mathbf{z} \sim p_\theta} \left[ ||\mathbf{z} - \epsilon||^2 \right]$, where $p_\theta(z) = \prod_{j=1}^{d} \Gamma(z_j; k_j, \mu_j)$, $\theta = \{k, \mu\}$, and $\epsilon = .49$. In this case, we only need to compute the first and second order derivatives of the function $f$.

**Toy problem 2**: $\mathcal{L}(\theta) = \mathbb{E}_{\mathbf{z} \sim p_\theta} \left[ \sum_{j=1}^{d} \exp(-\epsilon \mathbf{z}_j) \right]$, in this case, the infinite sum transfers to $\epsilon$, which results in the following estimator: $\nabla_\theta \mathcal{L} = \nabla_\theta \log \varphi_\theta \left( \frac{\epsilon}{i} \right) \mathbb{E}_{\mathbf{z} \sim p_\theta} \left[ f(\mathbf{z}) \right]$.

In figures 1 and 2 we report the training loss and log variance of the gradient across iterations of gradient descent for different values of the dimension $d \in \{1, 10, 100\}$. The stochastic backpropagation estimator converges to the minimal value in all cases faster than the other estimators and the variance of the gradient is competitive with the pathwise gradient.

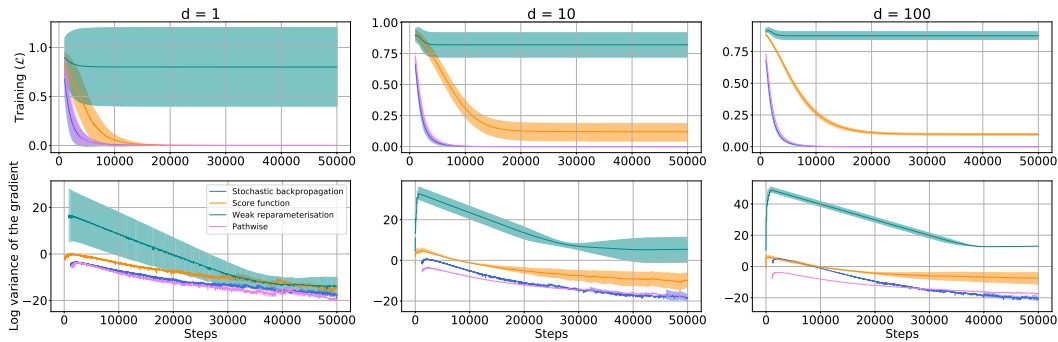

Figure 2: Training loss and log variance of the gradients for the different estimators for $f(z) = \sum_{j=1}^{d} \exp(-\epsilon z_j)$ for $d \in \{1, 10, 100\}$.

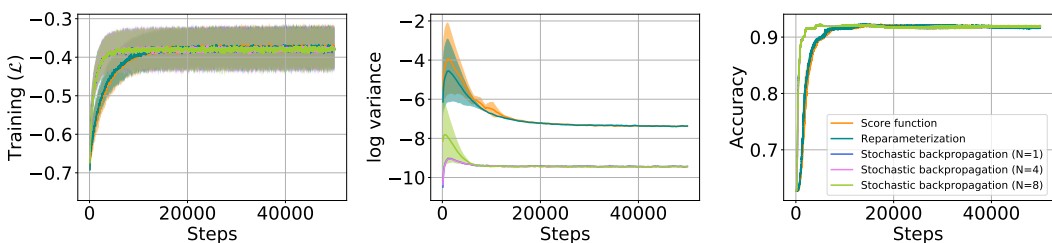

Figure 3: Bayesian Logistic Regression with Laplacian priors

## 6.2 BAYESIAN LOGISTIC REGRESSION WITH LAPLACIAN PRIORS

We evaluate the Laplace stochastic backpropagation estimator using a Bayesian logistic regression model (Jaakkola & Jordan, 1997), similarly to (Mohamed et al., 2019). In our case, we substitute the normal prior and posterior on the weights with Laplace priors and posteriors. We adopt the same notations of (Murphy, 2012), where the data, target and weight variables are respectively: $x_n \in \mathbb{R}^d$, $y_n \in \{-1, 1\}$, and $\mathbf{w}$. The probabilistic model in our case is the following:

$$p(w) = \prod_{j=1}^{d} L(w_j, 0, 1) \qquad p(y|\mathbf{x}, \mathbf{w}) = \sigma(y\mathbf{x}^T\mathbf{w}), \tag{25}$$

where $\sigma$ represents the sigmoid function. We consider Laplacian variational posteriors of the form $p_\theta(w) = \prod_{j=1}^{d} L(w_j, \mu_j, b_j)$, with $\theta = \{\mu, b\}$. The evidence lower bound of a single sample is given by:

$$\mathcal{L}(x_n, y_n; \theta) = \mathbb{E}_{\mathbf{w} \sim p_\theta} \left[ \log \sigma(y_n x_n^T \mathbf{w}) \right] - \mathbb{D}_{KL}[p_\theta||p], \tag{26}$$

where the Kullback-Leibler divergence between the two Laplace distributions is the following:

$$\mathbb{D}_{KL}[p_\theta||p] = \sum_{j=1}^{d} \left\{ |\mu_j| + b_j e^{-\frac{|\mu_j|}{b_j}} - \log b_j - 1 \right\}. \tag{27}$$

We test the model on the UCI women's breast cancer dataset (Dua & Graff, 2017), with a batch size of 64 and 50 samples from the posterior to evaluate the expectation. In the case of the stochastic backpropagation estimator we truncate the infinite series for the scale parameter $b$ of equation 19 to $N = 4$ and $N = 8$. In figure 3, we report the training evidence lower bound, the log variance of the gradient, and the accuracy computed on the entire dataset for the different estimators. The stochastic backpropagation estimator converges faster than the considered estimators and the variance is significantly lower. We also notice that the truncation level of the infinite series for the scale parameter has little effect on the outcome. In figure 4, we report the bias and variance of the estimator at different values of the truncation level, for a fixed parameter value during the training phase (epoch=100). The bias and variance do not vary much, with the truncation level in this case, this

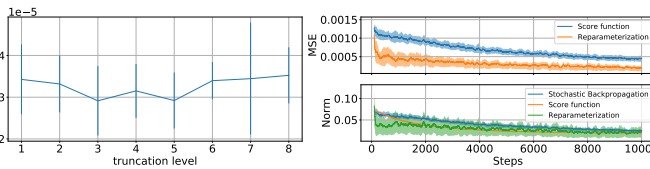

Figure 4: (Left) Bias and variance of the gradient for different values of the truncation level at a fixed parameter value. (Right-top) Mean square error between the Laplace gradient estimator and the score function and reparameterization estimators across iterations. (Right-bottom) Norm of the gradient estimators.

result confirms the intuition of neglecting higher frequencies presented in section 5. In addition, we compare the mean squared error between the Laplace stochastic backpropagation estimator and the score function and reparameterization estimators. As shown in figure 4 the mean squared error is small, thus the values of the gradients across iterations are close. However, the reparameterization gradient is closer to our estimator than the score function gradient, probably due to the fact that the reparameterization gradient is more stable and has lower variance.

## 7    RELATED WORK & DISCUSSION

Computing gradients through stochastic computation graphs has received considerable attention from the community, due to its application in many fields. The first general approach that provides a closed form solution for any probability distribution is the score function method (Glynn, 1989; Williams, 1992; Sutton et al., 2000; Schulman et al., 2015). The main inconvenience of this approach, is that it results in high variance gradients when the dimension of the random variable becomes high. In order to bypass this issue, the second approach consisted of designing control variates to reduce the variance of the score function estimator (Paisley et al., 2012; Weaver & Tao, 2013; Mnih & Gregor, 2014; Ranganath et al., 2014; Tokui & Sato, 2017). In addition to the score function gradient, it was proposed to use an importance weighted estimator instead of the classical score function with a multi-sample objective (Mnih & Rezende, 2016; Burda et al., 2015).

The second class of approaches is that concerning reparameterization tricks (Kingma & Welling, 2013; Rezende et al., 2014). Through the decoupling of the computation of the gradient from the expectancy, reparameterization tricks have shown that they provide low-variance gradients using often a single sample. The issue for these methods is the necessity to find a reparameterization for each probability distribution. Certain distributions such as the Gaussian are easy to reparametrize but others like the gamma are not. In addition, discrete random variables do not admit an easy reparameterization as well. Recently, these issues has been partially solved through implicit reparameterization, the generalized reparameterization gradient, and the pathwise gradient (Ruiz et al., 2016; Figurnov et al., 2018; Jankowiak & Obermeyer, 2018). For the discrete case, continuous relaxations that are reparameterizable have been proposed and combined with control variate methods (Gu et al., 2016; Maddison et al., 2016; Jang et al., 2016; Tucker et al., 2017; Grathwohl et al., 2018).

Our approach, in contrast provides a new broad family of stochastic backpropagation rules derived using the Fourier transform. One interesting aspect of our approach is the fact that the weighting $a_n$ is separated from the expectation of the higher order derivatives of the function $f$. Thus the sampled variable does not intervene in the weighting in contrast to other methods such as reparameterization and pathwise gradients. In addition, if the function $f$ contains weakly correlated terms, by applying the derivative, some random variables would be eliminated. Thus the estimators of the higher order derivatives would be sampled with respect to the derivation variable (and other correlated variables), which would result in lower variance.

It is worth noting that deriving stochastic backpropagation rules using the Fourier transform has been proposed for the Gaussian case (Fellows et al., 2018). In our work, we extend it to non Gaussian distributions by way of the characteristic function, and exploiting the invariance of the functional inner product under Fourier transformation (Parseval's theorem).

## 8 CONCLUSION

In conclusion, in this paper we presented a new method to compute gradients through random variables for any probability distribution, by explicitly transferring the derivative to the random variable using the Fourier transform. Our approach, gives a framework to be applied for any distribution, where the gradient of the log characteristic function is analytic, resulting in a new broad family of stochastic backpropagation rules, that are unique for each distribution.

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

| Dataset | Model | REBAR | RELAX | BSB (S=1) | BSB (S=5) | BSB (S=10) |
|---|---|---|---|---|---|---|
| MNIST | one layer SBN | -114.14 ± 0.44 | -114.55 ± 0.48 | -110.87 ± 0.2 | -110.70 ± 0.11 | **-110.59 ± 0.08** |
| | two layer SBN | -101.33 ± 0.04 | -101.09 ± 0.07 | **-99.74 ± 0.3** | -100.44 ± 0.28 | -100.66 ± 0.21 |
| | Bern. VAE | -127.76 ± 0.84 | -128.06 ± 2.66 | **-107.4 ± 1.47** | -108.46 ± 0.37 | -109.19 ± 1.31 |
| Omniglot | one layer SBN | -123.66 ± 0.05 | -123.82 ± 0.17 | **-113.53 ± 0.21** | -114.34 ± 0.16 | -114.37 ± 0.19 |
| | two layer SBN | -117.81 ± 0.17 | -117.89 ± 0.04 | **-102.05 ± 0.19** | -102.16 ± 0.09 | -102.29 ± 0.14 |
| | Bern. VAE | -136.83 ± 0.31 | -136.53 ± 0.32 | **-126.94 ± 0.81** | -128.69 ± 0.34 | -129.48 ± 0.38 |

Table 1: Test likelihood for the Bernoulli stochastic backpropagation (BSB) estimator compared to the REBAR and RELAX estimators. We report the mean and standard deviation over 5 runs.

## A  THE DIRAC DISTRIBUTION: THE LINK BETWEEN NEURAL NETWORKS AND PROBABILISTIC GRAPHICAL MODELS

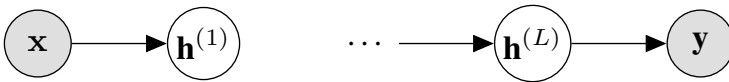

Figure 5: A hidden variable probabilistic model, where the observed variables are the data $\mathbf{x}$ and target $\mathbf{y}$, with $L$ hidden stochastic layers $\mathbf{h}^{(1:L)}$.

In this section, we explore the connection between neural networks and probabilistic graphical models following from the stochastic backpropagation rule of the Dirac delta distribution. To this end, let us consider the probabilistic graphical model of figure 5. The observed random variables in this model are denoted $\mathbf{x}$ and $\mathbf{y}$ representing the data and target variables. We place the analysis in a supervised learning context, but the argument is valid for unsupervised models as well. As usual the goal is to maximize the log likelihood for the data samples $(x, y)$, which is intractable, given that we need to integrate over the hidden variables. However using variational inference, we can maximize an evidence lower bound of the form:

$$\mathcal{L}(\theta; x, y) = \mathbb{E}_{\mathbf{h}^{(1:L)} \sim q_\theta(\cdot|x)} \left[ \log p(y, \mathbf{h}^{(1:L)}, x) \right] + \mathbb{H}[q_\theta(\cdot|x)] \tag{28}$$

As suggested in the Dirac stochastic backpropagation rule, let us assume that the variational posteriors and priors are Dirac delta distribution of the form:

$$q_\theta(\mathbf{h}^{(l+1)}|\mathbf{h}^{(l)}) = p(\mathbf{h}^{(l+1)}|\mathbf{h}^{(l)}) = \delta_{a^{(l+1)}(W^{(l+1)T}\mathbf{h}^{(l)}+b^{(l)})}(\mathbf{h}^{(l+1)}) \qquad \forall 0 \leq l \leq L-1 \tag{29}$$

where, the $a^{(l)}$, $W^{(l)}$, and $b^{(l)}$ represent respectively the activation functions, the weights and biases for layer $l$, with the convention $x := \mathbf{h}^{(0)}$. Under these assumptions, the Kullback-Leibler divergence term is equal to zero, and the evidence lower bound reduces to the the log-likelihood of a classical neural network:

$$\mathcal{L}(\theta; x, y) = \log p(y|g_\theta(x)), \quad \text{with,} \quad g_\theta(x) = a^{(L)}(W^{(L)}(....a^{(1)}(W^{(1)}x + b^{(1)})...)$$

Thus, when using neural networks we are indirectly using a probabilistic graphical model and making the strong assumption that the hidden layers follow a parameterized Dirac distribution knowing the previous layer.

## B  EXPERIMENTS USING DISCRETE STOCHASTIC BACKPROPAGATION

We evaluate the Bernoulli and Categorical Stochastic Backpropagation estimators (BSB and CSB) of equations 17 and 18 on standard generative modeling benchmark tasks, using the MNIST and Omniglot datasets (LeCun & Cortes, 2010; Lake et al., 2013). We use the REBAR, RELAX, and Gumbel-softmax (or Concrete) estimators as baselines for our comparison (Jang et al., 2016; Maddison et al., 2016; Tucker et al., 2017; Grathwohl et al., 2018). The Bernoulli stochastic backpropagation is compared to the REBAR and RELAX estimators for three models: the sigmoid belief network of one and two stochastic hidden layers (Neal, 1992) and the variational autoencoder.

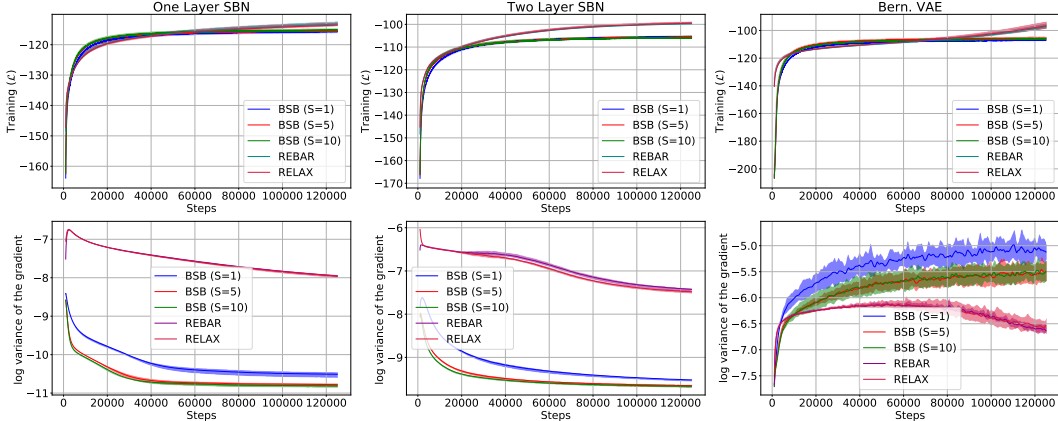

Figure 6: The training evidence lower bound on the MNIST training set (top) and the log variance of the gradient (bottom) over 5 runs. Comparison with the REBAR and RELAX estimators.

| Dataset | Model | Gumbel-softmax | CSB (S=1) | CSB (S=5) | CSB (S=10) |
|---------|-------|----------------|-----------|-----------|------------|
| MNIST | one layer | $-113.46 \pm 0.59$ | $-107.48 \pm 0.37$ | $-107.24 \pm 0.31$ | $\mathbf{-107.33 \pm 0.13}$ |
|  | Cat. VAE | $-122.97 \pm 5.68$ | $-103.49 \pm 0.73$ | $-102.68 \pm 0.63$ | $\mathbf{-101.78 \pm 0.88}$ |
| Omniglot | one layer | $-125.76 \pm 0.24$ | $-122.49 \pm 0.80$ | $-122.98 \pm 0.30$ | $\mathbf{-122.98 \pm 0.21}$ |
|  | Cat. VAE | $-140.25 \pm 1.99$ | $\mathbf{-130.20 \pm 0.74}$ | $-131.66 \pm 0.84$ | $-131.63 \pm 1.05$ |

Table 2: Test likelihood for the categorical stochastic backpropagation (CSB) estimator, compared to the Gumbel-softmax estimator. We report the mean and standard deviation over 5 runs.

In this case, we adopt the same architectures as (Grathwohl et al., 2018). The categorical stochastic backpropagation estimator is compared to the Gumbel-softmax estimator (Maddison et al., 2016; Jang et al., 2016) using two models: a variational autoencoder and a single layer belief network with categorical priors. In this case, we set the dimension of the hidden layer to $d = 20$ and the number of modalities for each dimension to $K = 10$.

All models are trained using the ADAM optimizer (Kingma & Ba, 2014) using a standard learning rate $\alpha = 10^{-4}$ and batch size of 100. We train the models for 500 epochs on the MNIST dataset and 100 epochs on the Omniglot dataset, longer learning epochs leads to overfitting and lower performance on the test sets for all estimators and models. We perform 5 iterations of training in all experiments and we report the mean and standard deviation of each performance metric considered.

For all models and estimators, we report the mean marginal test likelihood in tables 1 and 2 for both datasets. The test likelihood is estimated via importance sampling using 200 samples from the variational posterior. In all cases the stochastic backpropagation estimator, a control variate free method outperforms the baselines. In the case of the one layer sigmoid belief network the BSB estimator exhibits an increase of performance of about 4 nats in the case of the MNIST dataset and 10 nats in the case of the Omniglot dataset. We also vary the number of samples used to estimate the expectation in the stochastic backpropagation rule $S \in \{1, 5, 10\}$. We notice that using a single sample estimate does not hurt performance and leads to a faster training process.

We estimate the mean variance of the gradients w.r.t the parameters of the models using exponential moving averages of the first and second moments computed by the ADAM optimizer. The BSB estimator significantly outperforms the REBAR, RELAX estimators in terms of variance reduction with a difference of about 2 nats in the case of sigmoid belief networks, and 1 nat in the case of the categorical variational autoencoder on the mnist dataset, leading to a more stable training process as shown in figures 6 and 7.

Finally, we evaluate the computational overhead of the categorical stochastic backpropagation estimator compared the Gumbel-softmax estimator. We compare the two estimators in terms of execution time of one epoch of training. The comparison is done using GPU implementations on

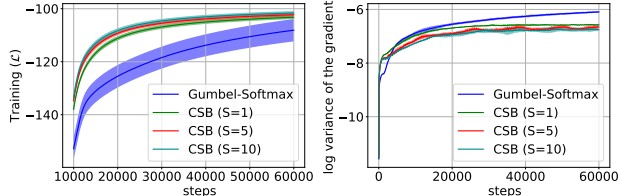

Figure 7: Training evidence lower bound and the log variance of the gradient for the categorical VAE on the MNIST dataset.

Table 3: Execution time of one epoch of training on the mnist dataset per estimator, per model.

| Model | GS | CSB (S=1) | CSB (S=5) | CSB (S=10) |
|---|---|---|---|---|
| one Linear layer | 4.11 (s) | 6.32 (s) | 10.93 (s) | 16.83 (s) |
| Cat. VAE | 4.13 (s) | 7.32 (s) | 13.29 (s) | 21.94 (s) |

a NVIDIA GeForce RTX 2080 Ti GPU, where the stochastic backpropagation rule of equation equation 18 is vectorized, thus leveraging the parallel batch treatment of the GPU. As shown in table 3, the Gumbel-softmax method is faster than stochastic backpropagation ($S = 1$) by a difference of about 3 seconds per training epoch. This is due to the forward passes performed to compute each of the terms in equation equation 18. The variance reduction and the increase in performance outweigh however the computational cost.

