# OpenReview forum: "Fourier Stochastic Backpropagation"
_ICLR.cc/2021/Conference — Reject_

### Official Review · AnonReviewer3 · 2020-10-26
**Major theoretical result in gradient estimation research. Possibly small errors, and discussion could be improved. Update: small errors were fixed, discussion was improved**

**Rating:** 10
**Confidence:** 4

**Review:**

**Update**
The errors in the paper were fixed, and the discussion was improved.
It is very rare to see a novel result as fundamental as the one presented in this paper, and I believe this puts it in the top 5% of accepted papers, so I have updated my score accordingly.

I think the discussion and experimentation still has room for improvement, but I am not too bothered, as there do not appear to be any major errors remaining in the paper.

**Summary**
The paper considers the task of estimating the gradient $\frac{d}{d\theta}\mathbb{E}_{p_\theta(z)}\left[f(z)\right]$.
This is a fundamental task relevant in all fields of machine learning, e.g. policy gradients, variational inference, or any other situation with stochastic computations.

The authors come up with a constructive method for deriving an estimator for this gradient for any distribution $p(z)$ based on Fourier analysis. The derived gradients include some known estimators, e.g. the Gaussian gradient identities, but also include some new estimators based on infinite series of higher order gradients of $f(z)$ for distributions such as Gamma or Laplace.

The analysis also works for Dirac delta distributions or discrete distributions.

They perform toy experiments to evaluate the new gradient estimators, and show that they work and seem to give better accuracy than previous pathwise gradients (but they use higher order gradients). For the infinite series based estimators, they truncated the length of the series at a certain depth (they tested up to 4th order and up to 8th order).

**Strengths**
Gradient estimation is fundamental, and deriving new methods is a great contribution.

The paper didn't discuss this well, but what is particular about the method derived is the form of the estimator based on separating out the computation of the expected gradients of $f(z)$, and the weighting applied to it, i.e. all of the gradient estimators derived with the method have the form $\sum_k a_k(\theta)\mathbb{E}_{p(z)}\left[\frac{d^k f(z)}{dz^k}\right]$. What is neat about this, is that one only needs estimators for the expected gradients $\frac{d^k f(z)}{dz^k}$, it is not necessary to know what the sampled $z$ values were. If you contrast this to some other gradient estimators, for example, the reparameterization gradient for the standard deviation parameter of a Gaussian distribution has the form $\epsilon \nabla_z f(z)$, where it is necessary to weight the separate values of $\nabla_z f(z)$ with different multipliers, but in the current paper, they derive a method where the multiplier is the same for all samples $z$. Moreover, from the derivation it is clear that there is only one way to perform this decomposition for each distribution, and the method to perform the decomposition for any $p(z)$ is provided in this paper. This is a general fundamental result.

**Weaknesses**

The discussion could be improved, e.g. the above points I mentioned were not explained.

I believe Lemma 3 and Corollary 3.1 are incorrect, because they ignore the singularity at the "kink" of the relu. This issue does not affect the experiments though, because relus were not used in the experiments. (This Lemma was about making the expressions based on infinite series of higher order derivatives tractable in cases where the higher order derivatives are 0.) I explain this in more detail later.

The discussion around the "discrete derivative" for categorical distributions was unconvincing to me. It appears that the derivation does not require selecting a special $z^{\*}$ choice, and the derivation goes through by just summing all $z$ without computing any difference at all.

From a practical point of view, probably the method will not be immediately used.

**Recommendation**

I recommend accepting the paper, because the theoretical result is fundamental. They performed experiments showing that the new methods work on toy problems, which I think is sufficient. Any issues I found were minor, and could be improved with a bit of revision. Probably, I will further increase my score if they adequately revise the paper.

**Questions**

In the experiments, what is the performance for truncation depth 1, 2, 3?
I am interested in at what point the performance starts degrading. Currently
the experiments only show 4 and 8, and both give similar performance, so it
is not clear whether, for example, 1 may also work or not.

In equation 6, $\alpha$ is a multi index, not an integer. What does
$(i\omega)^{\alpha}$ mean?

**Additional comments, suggestions, clarifications**
Kink in the relu: For example, consider $f(z) = max(0, z)$ is a relu. Then consider
$\mathbb{E}_{p(z)}\left[\frac{d^2 f}{dz^2}\right]$. Due to the effectively infinite second derivative at 0, I believe this expectation should be considered as $p(0)(\frac{df(0+\epsilon)}{dz} - \frac{df(0-\epsilon)}{dz}) = p(0)$, and I believe the values at the singularities will matter for the gradient estimator. As long as I have not misunderstood something, I would suggest to just remove the discussion about relus, and say that if the higher order derivatives disappear, it becomes tractable.

In general, I think the discussion would be greatly improved if you emphasize the structure of the gradient estimator based on separating out the weighting for the expected gradients $a$, and computing the expected gradient
$\mathbb{E}_{p(z)}\left[\frac{d^n f}{dz^n}\right]$. And contrast this to the other existing gradient estimators, e.g. reparameterization or pathwise derivatives, which often apply a different weighting for derivatives at different $z$ values. It would be good to emphasize that what you have derived is not a general form for gradient estimators, it is a particular (fairly broad) family of gradient estimators among other techniques that are not described by your derivation. I also did not find the discussion about the Dirac delta or the discrete distributions particularly insightful, so I would suggest to spend less time emphasizing these points.

"Furthermore, we show that the classical deterministic
backproapagation rule is a special case of stochastic backpropagation
where the distribution is a Dirac delta, bridging the domains of neural
networks and probabilistic graphical models."
I would not emphasize this point, as it is probably obvious to many
researchers. The Dirac delta can be acquired by letting $\sigma \to 0$
for a Gaussian distribution, which directly gives the result.

In the introduction, there are some spaces before the '?' signs, which should
be removed.

I disagree with some of the discussion in the introduction. In particular
the questions posed are already answered to some extent:

"How to develop stochastic backpropagation rules similar to those of
(Rezende et al., 2014) for a broader range of distributions?"
Implicit reparameterization gradients work for a broad range of
distributions. Moreover, the reparameterization based on the cdf of
a distribution always works as long as the cdf can be inverted (this is
just a computational issue, rather than a theoretical one).

"What is the link between the discrete random variable case and the
continuous case? And finally, what is the relation between stochastic
backpropagation and classical deterministic backpropagation?"
I wouldn't say the paper gives a definite answer to these
questions. It just provides another interpretation based on Fourier
transforms/characteristic functions, which is good, but I don't
believe the new interpretation is better than previous ones; it is
just different. I would tone down the discussion and just say that
you provide interpretations based on Fourier analysis.

sommable -> summable

Perhaps the multi-index notation could be clarified by 1-2 examples, and/or
emphasized by putting in a definition block. I am not sure whether it will
improve it though.

The weighting function for the derivatives is outside the expectation,
i.e. the weights do not depend on the sampled z position, and the z
only comes into play for computing the expectation of some gradients
of f(z). I think this structure of the gradient estimator should be
emphasized more.

Equation 3 claims that the weights are unique, so there is only one
way to construct the estimators that they have constructed. I think
this should be emphasized more. Also, instead of $\exists!$ it may be
better to just write "there exists a unique".

$a_\alpha$ is a bit difficult to read. Consider using different notation
for either $a$ or $\alpha$.

" Putting everything together": you have an extra space " " in the beginning
of the sentence.

In the Theorem 1 statement , can you write out that $a$ are the
Taylor weights of $\nabla_\theta\log \varphi_\theta(\omega)$?

"Plugging this expression to equation equation 5" -> there's a duplicate
"equation"

The "(AUEB & Lazaro-Gredilla,2015)" citation author names should be fixed
(the default on google scholar is not good).

The discrete derivative explanation in equations 10 and 13 is not
convincing to me. Instead of pulling out the $\varphi(z\*)$ term in eq
11, you could have just kept the summation across all $z$, and it
would have lead to the gradient estimator also summing all $f(z)$. It
is not clear why the difference between $f(z)$ and $f(z\*)$ is necessary
or why it should be interpreted as a discrete derivative as the $z$ vector
is not being perturbed.

Laplace is reparameterizable. What is meant by the pathwise derivative
in the experiments? The work from Jankowiak and Obermeyer defines a
family of gradient estimators not a single one.

For the toy problems. Rather than showing only the learning curves,
it would be good to also test that the expected gradients for the
new methods and old methods are the same at some particular parameter
values (up to estimation accuracy, but you should be able to get the estimates accurate by repeating the computation many times and averaging). Also, for the experiments with truncation, probably what is more important than the variance is the gradient accuracy. So, it would be good to show both the bias and variance at different truncation depths. (Ideally, I envision a graph showing bias and variance of the gradient at a particular parameter value plotted against different truncation depth values.)

Rezende et al (2014) also mention the reparameterization of the variable
under stochastic backpropagation, and this gives a different gradient
for the covariance parameters of a Gaussian compared to the stochastic
backpropagation rule derived in the present paper. Hence, it does not
generalize stochastic backpropagation as envisioned by Rezende. Instead,
it is a separate method for deriving one particular type of stochastic
backpropagation rule for any distribution. I would suggest something
like Fourier stochastic backpropagation, characteristic stochastic
backpropagation, Fourier expectation gradients, etc.

The log characteristic function for Laplace should be
$i\omega\mu - \log(1 + b^2\omega^2)$, you're missing the $\mu$ in
your equation.

"Our approach, in contrast generalizes stochastic backpropagation as
presented by (Rezende et al.,2014), where the derivative is explicitly
transported to the random variable"
No, it doesn't generalize it as presented by Rezende. In Rezende's work,
reparameterization is a subset of stochastic backpropagation (it is
listed under section 3 titled stochastic backpropagation). And
reparameterization contains gradient estimators not derived by your
method (e.g. the $\frac{d}{d\sigma}(\cdot)$ gradient for a Gaussian).
Hence, it is wrong to say that your method generalizes
stochastic backpropagation.

---

> ### Author Response · Authors · 2020-11-13
> **Response to reviewer 3**
>
> Thank you for acknowledging our novel contributions and for your valuable comments and suggestions to improve the paper.
>
> Response to questions:
> * Q.1 On the truncation and accuracy:
> Thank you for the suggestion, we have added the desired graphs. It seems that truncation N=1 works as well as N=4, N=8 leads to higher variance at early stages. At a fixed parameter value the variance and bias do not vary much. The MSE w.r.t older methods is small.
> * Q.2 meaning of $(i\omega)^\alpha $:
> We use here multi index notation $(i\omega)^\alpha = (i\omega_1)^{\alpha_1}(i\omega_2)^{\alpha_2}... (i\omega_d)^{\alpha_d}$.
>  * Q.3 Laplace is reparameterizable. What is meant by the pathwise derivative in the experiments?
> It is the reparameterization of the laplace from the uniform distribution that is used: $ \mu - b sign(u) \log(1 -  |u|) $. Thank you we have corrected this.
>
> Response to comments:
> * On the kink of the relu:
> Thank you for pointing this out indeed it is tricky, but the lemma is correct. Indeed at the singularity we have $ p(0) \lim_{\epsilon \to 0} \frac{\frac{df(0+\epsilon)}{dz} - \frac{df(0-\epsilon)}{dz}}{2 \epsilon} =p(0) \lim_{\epsilon \to 0} \frac{1}{2 \epsilon}  = \infty$. However, this is for the singularity at the specific point $0$. When considering the integral, one can write: $ \mathbb{E}_{p(z)} [\frac{d^2 f}{dz^2}] = \cancelto{0}{\int _{- \infty}^{ - \epsilon} p(z) \frac{d^2 f}{dz^2} dz}  +  \int _{- \epsilon}^{ \epsilon} p(z) \frac{d^2 f}{dz^2} dz  +  \cancelto{0}{\int _{\epsilon}^{\infty} p(z) \frac{d^2 f}{dz^2} dz} \xrightarrow[\epsilon \to 0]{}  \int _{0}^{ 0} p(z) \frac{d^2 f}{dz^2} dz  = 0$  because it is a point integral. This can also be justified in another way, the space of functions we are working on is $L^2$ which is the class of equivalences in which  two functions represent the same $L^2$ function if the set where they differ has measure zero [1]. Thus the equality in lemma 3. $ \partial _{z_j}^{\alpha_j} (\nabla_z g_\phi) = 0 $ is in the sense almost everywhere. The singularity points have zero measure thus we can simply ignore them, and formally by the previous decomposition we can show they do not change the value of the integral. Thank you for pointing this out, this indeed was a point that needed to be clarified, we have changed the lemma and proof to point out that the equality is in the sense almost everywhere.
> * Discussion on the separation of the weighting and expectation:
> Thank you for the suggestion, we had not picked up on this aspect, so we have updated the discussion in the paper  to include it.
> * "what you have derived is not a general form for gradient estimators, it is a particular (fairly broad) family of gradient estimators ":
> Indeed stochastic backpropagation does encompass other methods that are not related to our approach. We have made the modification in the introduction, discussion and conclusion of the paper to correct this aspect.
> * Connection to Rezende et al, and the term Generalized stochastic backpropagation:
> We made the claim that it generalizes rezende et al, mainly because the Gaussian case reduces to equation 9 in their paper, which they derived using integration by parts. Indeed, they take their analysis further and they derive the reparameterization trick which is now a subset of stochastic backpropagation methods. We have updated the introduction to eliminate this claim, we simply state that it reduces to rezende et al in the Gaussian case.
> * " I would suggest something like Fourier stochastic backpropagation, characteristic stochastic backpropagation, Fourier expectation gradients":
> Thank you for the suggestions, we will check if a modification of the title is allowed by ICLR at this stage of submission. If yes, something like Fourier Stochastic Backpropagation is indeed more on point.
> * " I would tone down the discussion and just say that you provide interpretations based on Fourier analysis." and the Dirac & Discrete connection:
> We have made the changes to highlight it as an interpretation, indeed it is obvious by taking the limit w.r.t $\sigma$, however we feel that probabilistic graphical models and neural nets are still quite separated in the literature, we felt it might be interesting to highlight the connection.
> As for the discrete case, we felt it is interesting because it can be casted as a discrete derivative [2]. And as shown in appendix B they perform as well or better than discrete relaxations and control variates. In fact, it seems that if you use the gradient in the form $ \sum_{k=1}^{K-1} \partial \pi_k Df(k)$ instead of $\sum_{k=1}^K \partial \pi_k f(k)$, we get lower variance and better elbo as the dimension $d$ grows, theoretically it's unclear why, thus we did not include it. 
>
>
> * Thank you very much for the corrections suggested, we have modified the manuscript to include them.
>
> References:
>  [1] : https://mathworld.wolfram.com/L2-Space.html
>  [2] : https://calculus.subwiki.org/wiki/discrete%20derivative

---

> > ### Comment · AnonReviewer3 · 2020-11-14
> > **On the kink of the relu**
> >
> > Thanks for the response, I will look at it after a bit. First I want to just respond that I still don't agree with the point on the kink of the relu.
> >
> > The integral over the singularity becomes:
> >
> > $\lim_{\epsilon\to 0} \int_{-\epsilon}^{\epsilon}p(z)\frac{d^2 f}{dz^2} dz = p(0)$
> >
> > To make the point clear, let's consider a simple analytic example with a Gaussian distribution $p(x;\mu,\sigma)$ where $\mu=0$ and $\sigma=1$, and let's consider the gradient w.r.t. $\sigma$, also let $f(z) = \max(0,z)$ is a ReLU.
> >
> > According to the Gaussian gradient identity, we have:
> >
> > $\frac{d}{d\sigma}\mathbb{E}[f(z)] = \frac{d\sigma^2}{d\sigma}\frac{d}{\sigma^2}\mathbb{E}[f(z)] = 2*\frac{1}{2}\mathbb{E}[\frac{d^2f}{dz^2}] = \mathbb{E}[\frac{d^2f}{dz^2}]$. According to your logic, this gradient will equal 0, because the second derivative is zero almost everywhere. But this is of course not correct.
> >
> > We can compute the gradient using the reparameterization trick:
> >
> > $\frac{d}{d\sigma}\mathbb{E}[f(z)] = \mathbb{E}[\epsilon\frac{df}{dz}] = \int_0^\infty p(\epsilon)\epsilon * 1 d\epsilon = \frac{1}{\sqrt{2\pi}} = p(0)$, which is the same result as I got before by taking into for the change in the gradient across the kink when considering $\mathbb{E}[\frac{d^2f}{dz^2}]$.

---

> > > ### Author Response · Authors · 2020-11-14
> > > **On the kink of the relu**
> > >
> > > Thank you for the clarification and explanation, indeed you are correct. Lemma 3 and corollary 3.1 do not hold we will make the suggested changes.

---

> > ### Comment · AnonReviewer3 · 2020-11-21
> > **Generally happy with modifications, a few more clarifications**
> >
> > In general, I like the paper even more now compared to before. I have a few more comments and questions below.
> >
> > One question:
> > 1. How were the bias and variance computed in Fig. 4? Also, for reference, it may be good to provide the gradient square magnitude, to show that the error between the methods is small compared to the gradient magnitude, and thus provide also computational proof that the methods give the same result.
> >
> > I would also like to clarify my previous comment about the derivation of the discrete case.
> > The derivation is correct as it is, but it seemed to me that it is needlessly complicated by picking out a $z*$ term, and I would like to ask for a comment on this; am I missing something? I show my derivation below:
> >
> > $\varphi_\theta(\omega) = \Pi_{c\in C} \varphi_\theta^{(c)}(\omega_c)$
> >
> > $\varphi_\theta^{(c)}(\omega_c) = \sum_{z_c} p_\theta(z_c)e^{i\omega_c^T z_c}$
> >
> > $\nabla_\theta \log \varphi_\theta^{(c)}(\omega_c) = \sum_{z_c} \frac{\nabla_\theta p_\theta(z_c)e^{i\omega_c^T z_c}}{\varphi_\theta^{(c)}(\omega_c)}$
> >
> > $\nabla_\theta \log \varphi_\theta(\omega) = \sum_{c \in C} \sum_{z_c} \frac{\nabla_\theta p_\theta(z_c)e^{i\omega_c^T z_c}}{\varphi_\theta^{(c)}(\omega_c)}$
> >
> > Plug into Eq. 6: $\int \hat{f}(\omega) \varphi_\omega (\omega) \sum_{c \in C} \sum_{z_c} \frac{\nabla_\theta p_\theta(z_c)e^{i\omega_c^T z_c}}{\varphi_\theta^{(c)}(\omega_c)} \mu(d\omega)$
> >
> > Rearrange:
> > $\sum_{c \in C} \sum_{z_c}\nabla_\theta p_\theta(z_c) \mathbb{E_{\mathrm{z_{-c}}}}\left[f(z_{-c}, z_c) \right]$
> >
> > It's the same as your derivation, except that I do not pull out the $z*$ term. So, it seems that it's a redundant step that unnecessarily increases the complexity. Do you have any comment on that?
> >
> > Also, it seems there's possibly an extra typo in your equations:
> > I believe equation 13 should have $\nabla_\theta \log \varphi_\theta^{(c)}(\omega_c)$ on the left-hand side.
> >
> > In the derivation that I provided, it turns into the same form as provided in the local expectation gradients paper. You mentioned that the form where you take the difference with the $z*$ term seems to give you better performance, but this seems very suspicious, so I think you are right in not including it without taking a very careful look at what is happening.
> >
> >
> > Some other comments I have (these are mostly suggestions, and it is up to you whether you want to include it):
> > Thanks for clarifying the Laplace pathwise gradient, but you should probably also mention in the experimental section in the paper what you mean by the pathwise gradient for Laplace.
> >
> > In terms of clarity, currently you have not explained reparameterization anywhere, yet in the discussion in the end you mention it. For better accessibility for newcomers to this research field, it may be useful to explain it with equations, and explicitly show an example where the weighting on the gradients can differ for different $z$.
> >
> > The multi-index notation is very important to understand for researchers to understand the paper, yet it is not necessarily a standard method in machine learning. So, I would recommend to emphasize the definition more, and spend a bit more time explaining it, e.g. showing simple examples with n = [1,0,0], n=[1,0,2] or something like that.
> >
> > For z, you sometimes boldfont it to indicate that it is a vector, other times it is a vector yet it is not in bold. There are also some other variables, such as $\omega$ that are vectors, yet not in bold. I would suggest to make this consistent. Perhaps the multi-index, n, could also be put into bold font, so that it would be more difficult to mistake it for a regular $n$-th order derivative. For example, the bm package in latex can be used to put Greek letters into bold.

---

> > > ### Author Response · Authors · 2020-11-22
> > > **Magnitude, discrete case and further revisions**
> > >
> > > Thank you very much for the additional suggestions.
> > >
> > > Response to questions:
> > > 1. The bias and variance are computed using the moving averages like in the Adam optimizer. We have added the magnitude of the gradients.
> > >
> > > Comment on the discrete case:
> > >
> > > Indeed your derivation is correct, and the specific choice of $z*$ can be set using any of the assignments. It does add a layer of complexity but we have an intuition why it works better. Let us consider the categorical case with d = 2 and K = 3 for example, and the different dimensions being independent (equation 18). The output neurons would be in this case $ \pi^{(1)}_1 , \pi^{(1)}_2, \pi^{(1)}_3, \pi^{(2)}_1 , \pi^{(2)}_2, \pi^{(2)}_3 $. Depending on which derivation we adopt we backpropagate different gradients.
> > > 1. In the difference case (discrete SB) the values backpropagated would be
> > > $ [f(z_1, 1) - f(z_1,3),  f(z_1, 2) - f(z_1,3), 0, f(1, z_2) - f(3,z_2), f(2, z_2) - f(3,z_2), 0]$
> > > 2. In the classical case without differences the values backpropagated are:
> > > $ [ f(z_1, 1) ,  f(z_1, 2) , f(z_1, 3), f(1, z_2), f(2, z_2),  f(3,z_2)]$
> > >
> > > So in the first case you would have d neurons with zero as the value backpropagated. in the other case the d neurons have the values $ f(z_{-j}, K)$ that could fluctuate and create variance. This is a conjecture that we intend to test, but when using differences it seems that the variance is significantly reduced.  Thus, theoretically the estimators are equivalent, our guess is that it is an implementation perk, because the gradient with differences has less freedom on d neurons than that without.
> > >
> > > Thank you for the extra suggestions and corrections. We wanted to indicate by bold font that the variable is a random variable, we corrected the zeta variables  to bold font in the Dirichlet and beta sections.  We will make the changes to include an example for the multi-index notation.

---

### Official Review · AnonReviewer4 · 2020-10-28
**The authors propose a general framework for deriving stochastic back-propagation rules, reconnecting with known rules and proposing a general method for deriving new ones.**

**Rating:** 6
**Confidence:** 3

**Review:**

In the present work, the authors present a general method for deriving stochastic back-propagation rules, using the link between Fourier transforms and the characteristic function associated to the random variable probability distribution and transferring the derivative directly to the random variable. They are thus able to encompass many well known back-propagation rules from the literature and derive new ones for different special cases.
The presented numerical experiments show how this method can be used to match state-of-art performance in simple models.
The authors also briefly discuss the bottlenecks associated to this method and propose some workarounds and rules of thumb (e.g., truncation of the series expansion entailed in the method) that seem to allow robust and efficient optimization.
Finally, the authors highlight the fact that deterministic neural networks and usual back-propagation can also be framed by the same method, by simply considering a Dirac's delta probability distribution for the parameters.
I think the paper is well written and that the presented framework nicely connects many ideas and methods developed in the literature in the past decades. It seems clear that methods based on the reparametrization trick will always be more viable for deep models, but they are based on ad hoc rules. The presented method is instead general and might be more effective in special cases where an early truncation of the series is justified. Personally, I would only put less accent on the link with deterministic back-propagation, which looks more like a sanity check than an important results.

---

> ### Author Response · Authors · 2020-11-13
> **Response to reviewer 4**
>
> Thank you for acknowledging our contribution and for your valuable comments.
>
> * "I would only put less accent on the link with deterministic back-propagation, which looks more like a sanity check than an important results."
> Indeed it can be seen as a sanity check. toning down the discussion on this aspect seems to be a shared suggestion with reviewer 3. We have updated the manuscript accordingly. Thank you

---

### Official Review · AnonReviewer2 · 2020-10-29
**A good finding but not enough for publishing as a conference paper**

**Rating:** 5
**Confidence:** 4

**Review:**

This paper presents a general framework for deriving stochastic back-propagation rules for both discrete and continuous distributions. With the help of characteristic function and Fourier transform, this paper derives a general formulation to calculate the gradients of model parameters with respect to the random variables.

The main contribution of this paper is to derive two general formula of gradient computation with respect to continuous and discrete random variables. The technique used in this paper is standard, and the applications on different distributions can conclude similar results as previous research works. It also verifies the correctness of the proposed generalized stochastic back-propagation. In general, this is a good finding or connection with the help of characteristic function and Fourier transformation. However, the content of this paper seems not enough to be a conference paper, and it is more like a workshop paper. In addition, using Fourier transformation (https://arxiv.org/pdf/1808.03953.pdf) in such case is not new. Another weakness I want to mention is the experiment conducted in this paper is not solid. Bayesian Logistic Regression is a quite toy setting and used in a bunch of variational inference papers. This paper should compare with some baselines. Similarly, the MNIST and Omniglot are two popular and widely used datasets. It should be easy to find reproducible baselines.

Some missing references:
[1] https://papers.nips.cc/paper/5670-fast-second-order-stochastic-backpropagation-for-variational-inference.pdf
[2] http://papers.nips.cc/paper/8325-provable-gradient-variance-guarantees-for-black-box-variational-inference.pdf

---

> ### Author Response · Authors · 2020-11-13
> **Response to reviewer 2**
>
> Thank you for the time accorded to reviewing our paper and for your comments.
>
> Response to comments:
>
> *  "The technique used in this paper is standard, and the applications on different distributions can conclude similar results as previous research works"
> Indeed the application for some known estimators coincide, however our approach can be applied to other distributions and we show that for the Laplace, gamma and Dirichlet distributions. And the recipe can be applied to any distribution with an analytic gradient of the log characteristic function. It could seem in some sense as a standard technique, however to the best of our knowledge, the method and estimators that we present are novel and not present in any previous work.
>
> * " In addition, using Fourier transformation (https://arxiv.org/pdf/1808.03953.pdf) in such case is not new"
> Indeed we have come across this paper while researching previous methods. But the scope of the analysis is unrelated and does not intersect with ours. The authors treat boolean functions and use discrete Fourier transforms, and the estimators found by their analysis do not coincide even with the Bernoulli case in our paper.
>
> * "Bayesian Logistic Regression  ... it should be easy to find reproducible baselines."
> Indeed the settings in which we test our approach are not sophisticated. This however is the case for two reasons: (i) The objective of the paper is to communicate the  theoretical results and simply back them up with experiments showing that the method works, which seems to us a standard practice used in several published papers in conferences; (ii) The complexity; unless we truncate the infinite series to level 1, we need to compute higher order derivatives of a neural network, which, with automatic differentiation is slow and presents numerical instabilities. Section 5 presents some cases where we can bypass these issues Detailed experimental evidence is still an open question that could be answered in future work.
>
> * On the missing references:
> https://papers.nips.cc/paper/5670-fast-second-order-stochastic-backpropagation-for-variational-inference.pdf : This reference treats second order methods, whereas our method is first order. Thus a discussion of this contribution could present some confusion for the reader.
> http://papers.nips.cc/paper/8325-provable-gradient-variance-guarantees-for-black-box-variational-inference.pdf: In our case we cannot use the variance analysis of this paper, as it only treats reparamaterizable location scale families. The distributions we treat are broader, the only variance analysis that we can realize is similar to that of (rezende et al) where we suppose weakly correlated terms. We added this aspect in the revised discussion phase as suggested by reviewer 3.

---

### Official Review · AnonReviewer1 · 2020-10-29
**Interesting approach with limitations**

**Rating:** 5
**Confidence:** 3

**Review:**

The paper derives a unified view for stochastic back-propagation for both continuous and discrete distributions. It uses the Taylor expansion of the characteristic function and then inverse Fourier transform to convert them infinite series of high order derivatives of the original function.

It is nice that the paper exposes several interesting connections between stochastic back-propagation in both continuous and discrete cases as well as deterministic back-propagation.

However, it seems the approach is quite limited in its usage and it is not so clear what the advantage is. Even though the paper addresses some issues with higher order derivatives and approximating the infinite Taylor series with truncation, it seems it is only applicable to a few distributions.

Although synthetic experiments show advantage in the proposed approach in some toy examples, it is not clear why the proposed approach is better than alternatives, in terms of lower variance or faster convergence.

Also it would be useful to fully specify one or two discrete examples and experiment with them in applications such as sparse feature selection etc.

---

> ### Author Response · Authors · 2020-11-13
> **Response to reviewer 1**
>
> Thank you for acknowledging our contribution and for your comments
>
> Response to comments:
> * "However, it seems the approach ... it seems it is only applicable to a few distributions."
> The approach can be used as long as the gradient of the log characteristic function is analytic, which is the case in many practical situations. For most used distributions, even complex ones such as von Mises, we can apply this analysis. We cannot claim that this approach can work with all distributions, If the reviewer could provide us with an example of a distribution for which the analysis would not work we would be grateful, and it could be an interesting case.
>
> * "Although synthetic experiments show ... lower variance or faster convergence."
> We gave an intuition of reduction of the variance in the discussion phase. If we suppose weakly correlated terms in the function f, applying derivatives eliminates terms depending on other random variables, thus the expectation would be only taken on the derivation variable, which results in lower variance.
>
> * "Also it would be useful to fully specify one or two discrete examples and experiment with them in applications such as sparse feature selection etc."
> We have included experiments on discrete examples in appendix B. We have tested sigmoid belief networks and variational autoencoders with discrete hidden variables (bernoulli and categorical). We show in this appendix B that Discrete stochastic backprop is competitive with state-of-the-art control variate methods. We did not include this in the paper, because the experiments coincide with other experiments in the litterature. Thus they might not be considered as a novel contribution.

---

### Decision · Program_Chairs · 2021-01-07
**Final Decision**

**Decision:**

Reject

**Comment:**

The focus of the paper is stochastic backpropagation for both continuous and discrete random variables. By using standard results from Fourier analysis the authors rewrite the corresponding gradients in an infinite weighted sum form ((3) and (9)), extending the results of (Rezende et al. 2014) and (Fellows. et al., 2018). The efficiency of the approach is illustrated in 2 toy examples.

As summarized by the reviewers, the problem tackled is interesting. However, they also pointed out that the novelty of the approach is quite limited and its practical usefulness is not clear (it should by demonstrated against state-of-the-art baselines, on realistic benchmarks).